# Antioxidant Properties of New Phenothiazine Derivatives

**DOI:** 10.3390/antiox11071371

**Published:** 2022-07-14

**Authors:** Olesya Voronova, Sergey Zhuravkov, Elena Korotkova, Anton Artamonov, Evgenii Plotnikov

**Affiliations:** 1School of Earth Sciences & Engineering, National Research Tomsk Polytechnic University, 30 Lenin Avenue, 634050 Tomsk, Russia; oaa@tpu.ru (O.V.); eikor@tpu.ru (E.K.); 2School of Nuclear Science & Engineering, National Research Tomsk Polytechnic University, 30 Lenin Avenue, 634050 Tomsk, Russia; zhuravkovsp@tpu.ru; 3Institute for Biomedical Problems, Russian Academy of Sciences, 76 A Khoroshevskoe, 123007 Moscow, Russia; anton.art.an@gmail.com; 4Research School of Chemistry & Applied Biomedical Sciences, National Research Tomsk Polytechnic University, 30 Lenin Avenue, 634050 Tomsk, Russia; 5Tomsk National Research Medical Center, Mental Health Research Institute, 4 Aleutskaya, 634014 Tomsk, Russia

**Keywords:** phenothiazine, phenothiazine derivatives, antioxidant activity, free radicals, voltammetry

## Abstract

Phenothiazine and its derivatives have a number of properties that contribute to their wider practical use in the production of biologically active substances, drugs, dyes, etc. Therefore, the synthesis and study of new compounds is of great relevance. The aim of this work was to investigate the antioxidant activity of a number of new phenothiazine derivatives. The patterns of electroreduction of oxygen and its radicals in the presence of phenothiazine derivatives in aqueous ethanol media were studied by voltammetry. The influence of various factors on antioxidant activity was considered by the methods of experiment planning. The optimal conditions for the manifestation of the antioxidant activity of phenothiazine derivatives have been found, which seems to be relevant since it opens up new possibilities for their further use as complex preparations with antioxidant activity, including in psychiatric practice.

## 1. Introduction

Phenothiazine and its derivatives are one of the most important and promising groups of chemical compounds that are widely used in various fields of chemistry and medicine. These compounds are used in dyes, inhibitors of the oxidation and polymerization of monomers in liquid organic media, to stabilize polymers of various classes, and even photosensitizers [1,2,3,4]. They are characterized by a wide spectrum of action on the human body. The widespread use of phenothiazine derivatives is due to the high biological and pharmacological activity, which depends on the chemical structure of the compounds. Phenothiazine derivatives have revolutionized the treatment of mental illness, opening the era of psychopharmacology with the use of chlorpromazine [5]. Despite the emergence of new antipsychotics, phenothiazine derivatives are of great clinical importance in psychiatry to date. In wide medical practice, phenothiazine derivatives are also successfully used as antihistamines [6]. In addition, their antibacterial, antitumor, and antituberculosis effects are known [7,8,9,10,11,12]. The synthesis of new phenothiazine derivatives is still ongoing. It should be noted that an important nonspecific pathological process in a number of diseases is oxidative stress [13,14,15,16,17]. In all aspects of application, interaction with oxygen and its radicals becomes a key point for novel phenothiazine complexes. Previously, antioxidant properties have been shown in various phenothiazine derivatives [18,19]. Some phenothiazine derivatives exhibited reducing power potential to convert Fe(3+) to Fe(2+) and high ability to scavenge H_2_O_2_ free radical in vitro. These activities were comparable to ascorbic acid as a standard antioxidant [20]. For example, phenothiazine derivatives inhibit the autoxidation of methyl linoleate [21]. Some authors have shown a prospective combination of antioxidant properties and biological activity [22]. Based on the literature data, we can predict high antioxidant properties in the compounds we have obtained. However, experimental confirmation of these properties and study of the mechanisms required in vitro tests. Further study will obviously include in vivo studies on biomodels of human diseases.

From this point of view, testing new compounds for their ability to neutralize free radicals is pathogenically justified. There are several methods for assessing the antioxidant and antiradical activity of substances [23,24,25,26,27]. We suppose that the methods mimicking the processes of oxygen reduction in living cells could most objectively reflect the behavior of an antioxidant in real objects [28,29,30]. We suggest here the optimal choice—the method of cathodic voltammetry, which is based on the electroreduction of oxygen, similar in stages to the processes in the mitochondria of a living cell. Electrochemical properties and patterns of electrode reactions for some phenothiazine derivatives have been established earlier [31]. Previously, we studied the antioxidant activity of some new compounds [32,33], and developed an effective method for assessing antioxidant properties [34]. Therefore, the main section of this work was the study of the antioxidant activity of new phenothiazine derivatives, which opens up new possibilities for their further use as antioxidants, including in the form of combined drugs. 

The chemical structure of this compound is based on a heterocyclic system consisting of a thiazine heterocycle with two benzene rings (Figure 1). In this connection, they are good electron donors and easily enter into reactions with oxidation. 

The main purpose of this work was to evaluate the antioxidant activity of phenothiazine and its derivatives using the method of cathodic voltammetry and to determine the optimal conditions for the measurement. The substances studied in this work potentially have high biological activity based on the chemical structure. However, this requires confirmation in experiments in vivo. This work allows us to consider these substances in a new way and significantly expand the potential of their use as antioxidants.

## 2. Materials and Methods

*Phenothiazine* powder (Sigma-Aldrich, Darmstadt, Germany) was used for the experiments. The rest of the investigated samples of substances were obtained for research by chemical synthesis according to the following method. 

### 2.1. Cis-10-Propenylphenothiazine (cis-10-PPT)

To a suspension of 25 g (0.125 mol) of phenothiazine and 28 g of KOH in 250 cm^3^ of dry dimethyl sulfoxide at 80 °C, 25 cm^3^ (0.29 mol) of allyl bromide was added (Bp = 71–71.3 °C). The reaction solution was kept at the same temperature and stirred for 3 h. After complete conversion of phenothiazine, the reaction mixture was poured into 2000 cm^3^ of distilled water. The oily product was extracted from the water surface with benzene (three portions of 100 cm^3^); then, the benzene extract was washed and dried. The solvent was evaporated and the remnants of the reaction mass were distilled on the unit using a fore vacuum pump (at a temperature of 160 °C/1–2 mm Hg). The obtained 10-propenylphenothiazine (10-PPT) in the form of an oily liquid, collected in receiver flasks, was dissolved in portions with warm ethanol. After that, the flask was transferred to re-crystallization at a negative temperature. Then, after 5–10 h, the state of the mother liquor was checked—crystals should have fallen out of it. The precipitated crystals were filtered off on a Schott filter with a porosity of 160 pores. Then, the crystals were dried at room temperature. In total, 24.2 g (80%) of Cys-10-propenylphenothiazine (Cis-10-PPT) was received in the form of white needles with m.p. = 34–34.5 °C [35].

Gross formula of the obtained compound: C_15_H_13_NS. 

Elemental analysis results:

Found, % (C 75.39; H 5.31; N 6.10; S 13.10).

Calculated, % (C 75.27; H 5.48; N 5.85; S 13.39).

In the IR spectrum of Cys-10-PPT, an intense band is observed in the region of 1657 cm^−1^, and the band in the region of 945 cm^−1^ is attributed to the stretching and bending vibrations of the –C=C-bond, respectively (740 cm^−1^ (1,2-substituted benzene rings), 1600, 1500 cm^−1^ C=C aromatic rings). In addition, there is no absorption band of the free N-H bond in the IR spectra.

NMR spectrum: (CCl_4_) δ ppm contains signals of the CH_3_ methyl group in the region of 1.55 ppm (3H, *J* = 7 Hz), a weakly split doublet in the region of 6.16 ppm. =C-H (1H, *J* = 7 Hz) and a multiplet of protons of aromatic rings in the region of 6.5–7.1 ppm (8H).

### 2.2. 2-Methyl-1-ethyl-3-(10-phenothiazinyl)-2,3-dihydro-1H-pyrido-[3,2,1-k,l]phenothiazine

#### Propenylphenothiazine Dimer (DPPT)

This substance was obtained by polymerization of 10-propenylphenothiazine at 20–25 °C in an organic solvent in the presence of a catalyst—boron trifluoride etherate in an amount of 3.6 × 10^−3^ mol per 1 mol of the starting monomer.

To do this, 2.4 g (0.01 M) of cis-10-propenylphenothiazine (cis-10-PPT) dissolved in 10 cm^3^ of an anhydrous organic solvent of ethyl acetate were loaded into the flask, 0.1 cm^3^ (0.00003 M) of BF_3_⋅OEt_2_ in ethyl acetate. The synthesis was carried out at 20–25 °C. until complete conversion of cis-10-PPT. The progress of the reaction was monitored chromatographically (TLC, Silufol, Kavalier, Czech Republic, eluents: hexane:ether = 6:1). At the same time, 25–30 min after the addition of the catalyst to the reaction solution in the volume of the reaction mass, the appearance of crystals of the substance was observed, which then formed a precipitate on the walls of the flask. Then, within 1.5 h, a noticeable increase in the volume of the precipitate was observed. At the end of the reaction, the catalyst was neutralized with a solution of KOH in methyl alcohol. The precipitate of the substance that accumulated during the synthesis was filtered off, washed, and dried. Then, 1.6 g (66.6%) of white small crystals were obtained with m.p. = 210–211C, readily soluble in chloroform, benzene, poorly soluble in ethyl acetate, acetone, hexane [36].

Gross formula of the obtained substance: C_30_H_26_N_2_S_2_.

Elemental analysis results:

Found, % (C 76.70; H 5.69; N 5.43; S 12.99).

Calculated, % (C 75.31; H 5.43; N 5.85; S 13.38).

IR spectrum (Thermo Fisher Scientific (USA), FTIR spectrometer Nicolet 5700, KBr) 760 cm^−1^ (1,2-substituted benzene rings), 790, 740 cm^−1^ (1, 2, 3 substituted benzene rings), 1600, 1500 cm^−1^ C=C aromatic rings). In addition, there is no absorption band of the free N-H bond in the IR spectra.

NMR spectrum: (CDCl_3_) δ ppm contains signals characteristic of the ethyl group: 1.02 (3H, t, *J* = 6 Hz), 1.28 (2H, d, *J* = 7 Hz), as well as signals in the regions 1.4–1.7 (2H, m), 2.4–2.85 (1H, m), 3.94–4.2 (1H, m), and 4.8–5.1 (1H, d. *J* = 11 Hz) and proton signals aromatic rings 6.5–7.2 (15H, m), M^+^ = 478. 

### 2.3. 2-Methyl-1-ethyl-1H-pyrido-[3,2,1-k,l]-phenothiazine

#### Pyridophenothiazine (PyrPT)

For synthesis, to a suspension of 1.0 g (2.09 × 10^−3^ mol) DPPT in 7.5 cm^3^ of dioxane were added at 100C in 0.24 cm^3^ (3.45 × 10^−3^ mol) of acetyl chloride and 0.34 cm^3^ (4.18 × 10^−3^ mol) of pyridine. The reaction solution was kept at the same temperature for 1 h. The reaction was carried out until complete conversion of the starting material. The progress of the reaction was monitored by thin layer chromatography. After completion of the reaction, the reaction mixture was cooled to room temperature. The precipitate formed was filtered off. The filtrate was poured into 50 cm^3^ of water, the precipitate formed was filtered off. The precipitates were washed successively with 50 and 30 cm^3^ of hexane. All washings and filtrates were combined. The organic phase was separated from the aqueous solution. The product was extracted from the aqueous solution with hexane. The extracts were dried. The solvent was evaporated, the precipitate was purified on a chromatographic column (silica gel sorbent, eluent—hexane). The resulting substance was recrystallized from ethanol. In total, 0.494 g (85%) of 2-methyl-1-ethyl-1H-pyrido[3,2,1-k,l]phenothiazine were obtained in the form of yellow crystals, m.p. = 70–71 °C [37].

Gross formula of the obtained compound: C_18_H_17_NS.

Elemental analysis results:

Found, %: C 77.41; H 6.13; N 5.00; S 11.46.

Calculated, %: C 77.34; H 6.09; N 5.01; S 11.48.

The IR spectrum of this compound indicates the absence of absorption of the free NH group and the presence of an absorption band of the olefin bond -1680 cm^−1^, there are also bands: 740 and 790 cm^−1^ (1,2,3 substituted benzene ring), 755 cm^−1^ (1,2 substituted benzene ring).

NMR spectrum: (CDCl_3_), δ, ppm: 1.0 (3H, t, *J* = 6 Hz, CH_2_CH_3_); 1.6 (2H, m, CHCH_2_Me), 2.0 (3H, s, =CCH_3_), 4.5 (1H, t *J* = 6 Hz, NCH), 6.1 (1H, s ArCH=) 6, 6–7.1 (7H, m HAr) phenothiazine ring protons.

Mass spectrum: M^+^ = 279.

### 2.4. Antioxidant Activity Measurement

The measurements were carried out using an automated voltametric analyzer “TA-2” (LLC “Tomanalit”, Tomsk, Russia). The electrochemical cell consisted of a glass cup with a background electrolyte and holder with mercury-film indicator electrode, silver chloride reference electrode, and a silver chloride auxiliary electrode. Water–ethanol solution of 0.1 N NaClO_4_ in various concentrations (10%, 30%, 50%, 70%, 85%) was used as a background solution.

The voltametric method for determining the antioxidant activity of phenothiazine and its derivatives consisted of recording voltammograms of cathodic oxygen reduction. We compared current pike with and without addition of test substances with different concentrations (0.01%, 0.1%, and 1.0%) to the background electrolyte solution in the potential range from 0.0 to −1.0 V to establish the relationship between the antioxidant activity of the studied substances and their concentration. The basic process here is oxygen electro-reduction (O_2_ ER); this process is identical to the oxygen reduction in the cells and tissues of the body. Notably, the mechanism of this process in monohydric alcohols is similar to that in water due to the identity of the structure of these media and the presence of hydrogen protons. The antioxidant activity of phenothiazine derivatives in such environment depends both on the concentration of the studied substances and on the water–ethanol ratio in the supporting electrolyte. The details of the voltametric method for measuring antioxidant activity is described previously in [34,38].

## 3. Results

### 3.1. Investigation of Oxygen Electroreduction in the Presence of Phenothiazine and Its Derivatives

The process of ER O_2_ on the surface of a mercury-film electrode proceeds according to the scheme:(1)$$O_{2}+e^{-}⇄{O_{2}}^{.-}$$
(2)$${O_{2}}^{.-}+H^{+}⇄{\mathrm{HO}}_{2}$$
(3)$$HO_{2}+H^{+}+e^{-}⇄H_{2}O_{2}$$
(4)$$H_{2}O_{2}+2H^{+}+2e^{-}⇄2H_{2}O$$

Since phenothiazine and its derivatives readily dissolve in aprotic solvents, we first considered their effect on the O_2_ ER process in the aprotic solvent DMF. Thus, the effect of their interaction only with oxygen and the radical anion was evaluated (stage 1 according to the scheme). For a more correct interpretation of the results for living systems, we decided to use an ethanol solution, the mechanism of oxygen reduction that is similar due to the presence of hydrogen protons (stages 1–4)

In both cases, the substances exhibited antioxidant activity, reducing the current of ER O_2_, which indicates their interaction with oxygen and its radicals (Figure 2).

We studied the effect of the potential sweep rate on the current and potential of the oxygen reduction signal in the range from 20 to 200 mV/s to reveal the mechanism of O_2_ ER in the presence of phenothiazine and its derivatives.

The peak current of the O_2_ electroreduction in the presence of phenothiazine increases and the peak potential shifts to negative values with an increase in the potential scan rate (Figure 3). We studied dependences of the O_2_ electrochemical reduction current on the sweep rate to the power (W^1/2^) in the presence of the substances. It allows to reveal the reversibility of the electrochemical signal of O_2_ reduction.

The current directly depends on the sweep rate to the power at ½ degree for reversible and irreversible processes. However, a quasi-reversible process does not follow a linear relationship. The obtained dependences of the oxygen electroreduction current vs. the square root of the potential scan rate are nonlinear. This indicates the quasi-reversibility of the O_2_ ER process in the presence of phenothiazine and its derivatives, as well as the O_2_ ER process itself (Figure 4).

One of the criteria for the presence of a subsequent chemical reaction for quasi-reversible electrode processes is the presence of a linear dependence of the signal current peak potential on the logarithm of the potential scan rate to the power of ½ (log W 1/2).

However, nonlinear dependence was observed (Figure 5). Similar dependences were observed not only at 85% ethanol in the background electrolyte, but also for other more dilute ethanol concentrations.

The nonlinear dependence in various ethanol solutions confirms the occurrence of a quasi-reversible process of oxygen electroreduction. The mechanism of the influence of a substance on this process is similar at different compositions of the background electrolyte. There is also a potential shift towards negative values and peak stretching with an increase in the concentration of the studied substances in the background electrolyte solution (Figure 6). According to the literature data, this indirectly indicates the interaction of these substances by the CE mechanism with molecular oxygen dissolved in the electrolyte or the CEC mechanism with the preceding and subsequent chemical reactions of the interaction of the antioxidant with molecular oxygen and its reduction products (Figure 7).

### 3.2. Determination of the Optimal Conditions for the Evaluation of Antioxidant Activity by the Experimental Design

Experimental planning methods were used to evaluate the optimal conditions for the antioxidant activity detection of the phenothiazine derivatives [28]. The antioxidant activity of phenothiazine derivatives depends both on the concentration and on the water–ethanol ratio in the supporting electrolyte. Here, we used a full factorial experiment to build a mathematical model of the O_2_ ER process in the presence of the substances under study, as well as assess its adequacy and assess the significance of the coefficients of the regression equation [39].

The variable factors were the concentration of the test substance (X1) and the concentration of ethanol (X2) in the supporting electrolyte. Since the oxygen concentration in the electrolyte with different ethanol concentrations is different, it is more correct to use the relative change in the O_2_ ER current when the test substance is added. It directly corresponds to the antioxidant activity of the substance. Therefore, as a response function (Y), we used the relative change in the oxygen electroreduction current, expressed as a percentage according to the formula:(5)$$K^{*}=100\%\left(1-\frac{Ii}{Io}\right)\cdot \frac{1}{t}$$
where *I_i_*—value of limiting current ER O_2_, mkA; *I_o_*—the value of the limiting current of ER O_2_ in the absence of a substance in solution, _MK_A; *t*—process time, min.

The main characteristics of the plan are presented in Table 1.

It was found that this model adequately describes the process for phenothiazine and its derivatives. All coefficients of the linear model are significant and have a plus sign, indicating that both factors have a positive effect on the response function. Moreover, the first factor (X1)—the concentration of the test substance—has the greatest influence. The effect of the interaction of factors is not significant. Thus, the equations of the mathematical model are as follows (Table 2).

The response surface type “stationary elevation” is observed for all substances. It appears with a smooth increase in the response function with changing factors. The maximum is not detected here. The surfaces reach a plateau with an increase in the concentration of the studied substances. However, it was inappropriate to conduct studies at higher concentrations of phenothiazine derivatives, since the studies were based on a therapeutic dose (maximum daily dosage) of known drugs based on phenothiazine derivatives. Use of these drugs in large doses can lead to poisoning or death, and it should be taken into account when predicting medical use and selecting concentrations for biological tests. We used the kinetic criterion to assess the antioxidant activity of phenothiazine and its derivatives. Measurements were conducted taking into account differences in the concentration of dissolved oxygen. This criterion reflects the amount of active oxygen radicals that reacted with the antioxidant per minute, *K* (µM/min):(6)$$K=\frac{C_{o_{2}}}{t}(1-\frac{I_{i}}{I_{o}})$$
where *C_O_*_2_—oxygen concentration in the initial solution without substance, µM; *I_i_*—value of the limiting current of O_2_ electroreduction, mkA; *I_o_*—the value of the limiting current of electroreduction of O_2_ in the absence of a substance in solution, mkA; *t*—process time, min.

Table 3, Table 4, Table 5 and Table 6 present the results of a quantitative assessment of the antioxidant activity of substances with different concentrations (0.01%, 0.1%, and 1%) in water–ethanol solutions with different concentrations (30%, 50%, 70%, 85%).

Antioxidant activity increases in the row: phenothiazine < pyridophenothiazine < cis-10-propenylphenothiazine < propenylphenothiazine dimer. Antioxidant activity generally follows changes in the concentration of substances. There is a dose-dependent, but nonlinear increase in activity with increasing concentration. In some cases, a change in the comparative activity in this series of compounds is noted at low concentrations. Here we revealed the highest antioxidant activity of the DPPT substance.

## 4. Discussion

The results confirmed that antioxidant activity increases with the concentration of the studied substances. However, it was found that this parameter is significantly affected by the water–ethanol ratio in the supporting electrolyte. It should be noted that the second factor (the ratio of solvents in the supporting electrolyte according to results of experiment planning) has the greatest influence on the parameter of antioxidant activity. The coefficients of antioxidant activity of the studied phenothiazine derivatives were calculated with correction of dissolved oxygen concentration. In the studied series, DPPT has the highest activity, and phenothiazine has the lowest activity. A possible explanation for the observed phenomenon is as follows. It is known that a feature of covalently bound sulfur in the structure of phenothiazine derivatives is high oxidizability. Oxidation processes can take place in several stages, and oxidation products are formed, including oxide and sulfur dioxide. We assume that the manifestation of propenylphenothiazine dimer with a significantly higher antioxidant activity compared to other derivatives is associated precisely with their molecular structure. As mentioned, the propenylphenothiazine dimer contains two fragments of the phenothiazine heterocycle, and pyridophenothiazine, which is its derivative, contains only one heterocycle. The observed dependence is associated with the chemical structure of thiazine heterocycle with two benzene rings. In this connection, these compounds are good electron donors and easily enter into reactions accompanied by oxidation. Notably, that the dual nature of phenothiazine exhibits both antioxidant and prooxidant properties, and this phenomenon is realized depending on the solvent or microenvironment in the biological object. Thus, in aqueous solutions, it exhibits mainly antioxidant properties, while in fats it behaves as a pro-oxidant with the formation of a phenothiazinyl radical [40]. In general, the antioxidant properties of phenothiazines are based on the chemical properties of the parent molecule, phenothiazine. However, the addition of functional groups significantly increases the antiradical properties of phenothiazine derivatives [20]. Possibly, antioxidant activity is enriched by the presence of electron donors such as sulfur as well as the richness of hydrogen in the additional benzene rings for substitution. However, some functional groups do not increase the activity of the molecule [41]. In addition, the presence of synergistic activity in combination with diphenylamine derivatives was noted, which is also an interesting phenomenon [42].

Evaluation of the influence of the phenothiazine derivatives on the reversibility of the O_2_ ER process showed the following. The dependence of the oxygen electroreduction current on the sweep rate to the power of ½ is nonlinear. This fact indicates the quasi-reversibility of the O_2_ ER process on the mercury-film electrode in the presence of the studied substances.

Several possible mechanisms of interaction of the studied phenothiazine derivatives with oxygen and its active forms are suggested. The interaction of tested substances with molecular oxygen and oxygen reduction products goes by the chemical-electrochemical CE mechanism. This conclusion is indirectly confirmed by the absence of a linear dependence of the peak potential on the logarithm of the sweep rate to the power of ½, as well as the shift of the potential towards negative values. At the same time, the CEC mechanism with the preceding and subsequent chemical reactions of the interaction of the antioxidant with molecular oxygen and the products of its reduction are not excluded.

To assess the potential use of these new antioxidants, a biological activity prediction was performed using the PASS online software. All tested substances are likely to have Glycosylphosphatidylinositol phospholipase D inhibitor properties. The probabilistic indicator of activity was Pa = 0.901 (cis-10-PPT); Pa = 0.915 (Phenothiazine); Pa = 0.571 (Propenylphenothiazine dimer (DPPT)); and Pa = 0.753 (Pyridophenothiazine (PyrPT)). Here, the prediction of the biological activity of a chemical compound is based on the assumption that the activity is directly related to the structure. The probability of the presence of activity is expressed by the parameter Pa (probability “to be active”) that estimates the chance that the studied compound is belonging to the sub-class of active compounds. Accordingly, the closer calculated Pa value is to one, the higher the predicted probability of the compound having a certain activity. Thus, the high antioxidant activity that we identified can be manifested at the biological level through a specific inhibitory effect on phospholipase. It could significantly increase the antioxidant effect at the cellular level. It is known that glycosylphosphatidylinositol-specific phospholipase D may play an important role in inflammation, because it can hydrolyze the glycosylphosphatidylinositol anchors of several inflammatory membrane proteins. Therefore, the potential effect of the studied substances as inhibitors of glycosylphosphatidylinositol phospholipase and their high chemical antioxidant properties allows us to consider the studied substances as promising drugs for the correction of oxidative stress in various pathologies.

## 5. Conclusions

We proved high antioxidant activity of tested phenothiazine derivatives. The dependence of antioxidant activity on the molecular structure of the substance and the influence of various factors on their antioxidant activity was shown. It was revealed that the ER O_2_ process is quasi-reversible in the presence of the studied substances. The interaction of these substances occurs presumably by the CE mechanism with molecular oxygen dissolved in the electrolyte or by the CEC mechanism with the preceding and subsequent chemical reactions of the interaction of the antioxidant with molecular oxygen and oxygen reduction products. The optimal conditions for the manifestation of the antioxidant activity of phenothiazine and its derivatives were found, which opens up new possibilities for their further use as antioxidants, including as combined drugs. Revealed antioxidant properties in combination of predicted biological activity make these substances promising for the correction of oxidative stress in the complex therapy of mental and cardiovascular diseases, where the role of oxidative damage has been proven.

## Figures and Tables

**Figure 1 antioxidants-11-01371-f001:**
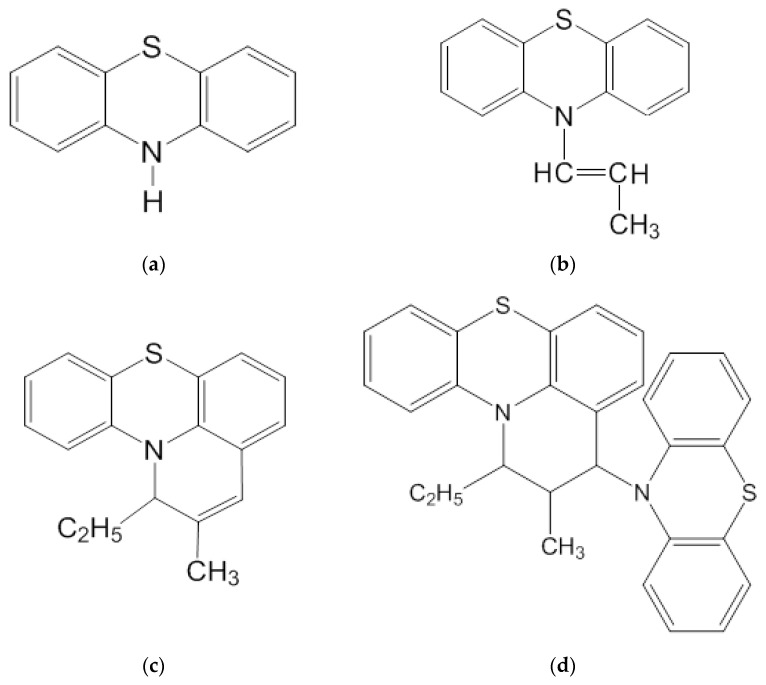
Structural formulas of phenothiazine and its derivatives: (**a**) phenothiazine (PT); (**b**) cis-10-propenylphenothiazine(cis-10-PPT); (**c**) 2-methyl-1-ethyl-1H-pyrido-[3,2,1-k,l]-phenothiazine. Pyridophenothiazine (PyrPT); (**d**) 2-methyl-1-ethyl-3-(10-phenothiazinyl)-2,3-dihydro-1H-pyrido-[3,2,1-k,l]phenothiazine. Propenylphenothiazine dimer (DPPT).

**Figure 2 antioxidants-11-01371-f002:**
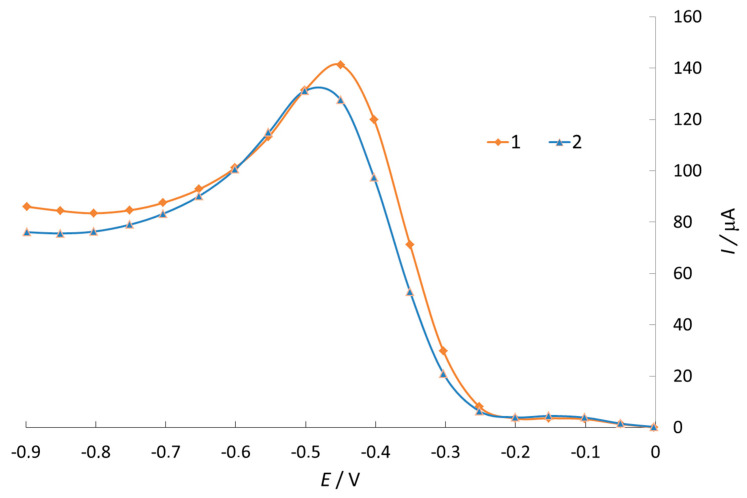
Voltammogram of ER O_2_ current 85% ethanol solution. 1—Background curve (0.1N NaClO_4_); 2—In the presence of 0.1% phenothiazine.

**Figure 3 antioxidants-11-01371-f003:**
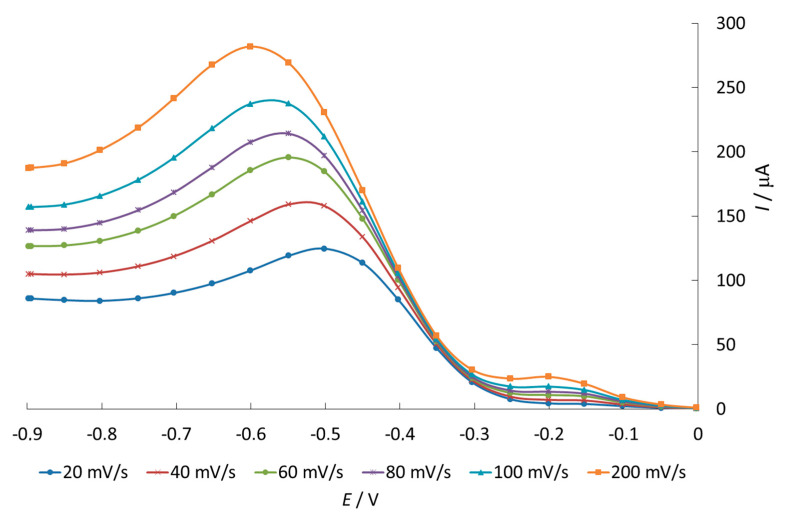
Voltammograms of O_2_ electroreduction on a mercury—film electrode versus the potential sweep rate in the presence of 0.1% phenothiazine in 85% ethanol solution.

**Figure 4 antioxidants-11-01371-f004:**
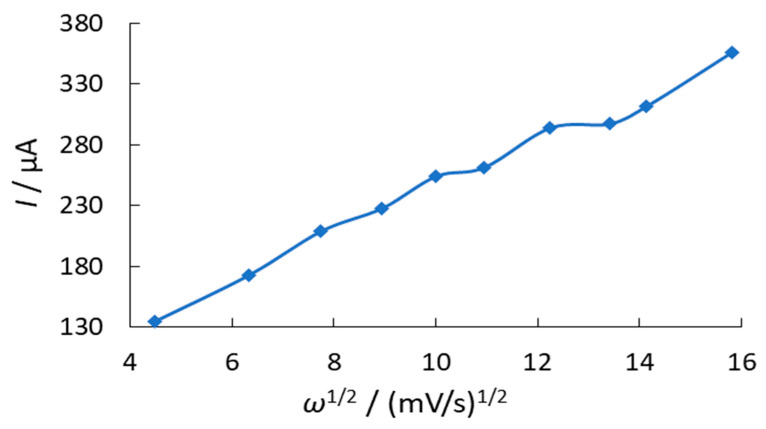
Dependence of the O_2_ electroreduction current on the RPE on the potential scan rate to the power of ½ in the presence of 0.1% phenothiazine in 85% ethanol solution.

**Figure 5 antioxidants-11-01371-f005:**
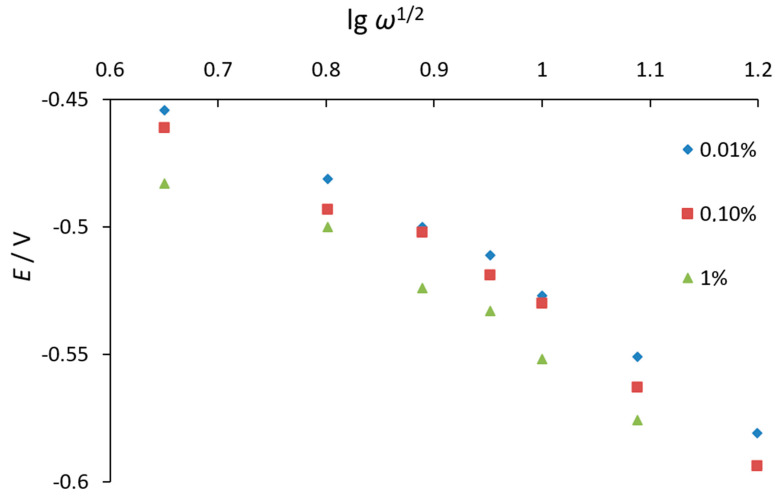
Dependence of the potential of the peak current of O_2_ electroreduction on a mercury–film electrode on the potential scan rate to the power of ½ in the presence of various concentrations of phenothiazine (0.01%, 0.1%, and 1%) in 85% ethanol solution.

**Figure 6 antioxidants-11-01371-f006:**
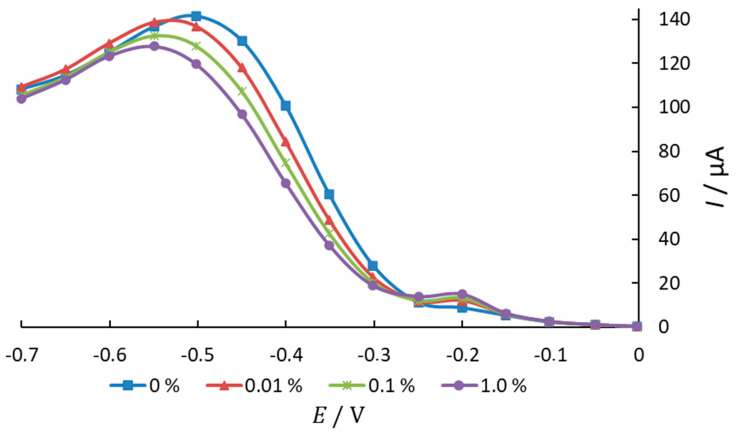
O_2_ electroreduction current voltammogram (mercury–film electrode), background curve (0.1N NaClO_4_, 85% ethanol solution), and in the presence of various concentrations of phenothiazine (0.01%, 0.1%, and 1%).

**Figure 7 antioxidants-11-01371-f007:**
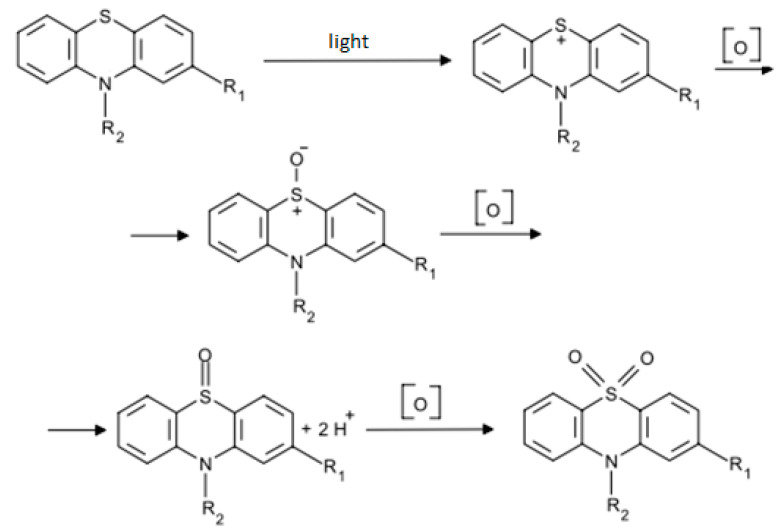
Preceding and subsequent chemical reactions of the interaction of the antioxidant with molecular oxygen.

**Table 1 antioxidants-11-01371-t001:** Main characteristics of the plan N = 2^2^.

Characteristic	X_1_ (C_AO_, %)	X_2_ (C_ethanol_, %)
Zero level	0.055	60
Variation interval	0.045	10
Upper level	0.1	70
Lower level	0.01	50

**Table 2 antioxidants-11-01371-t002:** The equations of the mathematical model.

Characteristic	Equations
PT	Y = 0.923 + 0.168 X_1_ + 0.098 X_2_
cis-10-PPT	Y = 0.46 + 0.17 X_1_ + 0.065 X_2_
PyrPT	Y = 0.23 + 0.138 X_1_ + 0.073 X_2_
DPPT	Y = 0.129 + 0.027 X_1_ + 0.022 X_2_

**Table 3 antioxidants-11-01371-t003:** Values of the antioxidant activity of phenothiazine and its derivatives in 30% water–ethanol solution.

30% Water–Ethanol Solution	C1 = 0.01%		C2 = 0.1%		C3 = 1%	
	K, µM/min	Sr	K, µM/min	Sr	K, µM/min	Sr
PT	1.44	0.01	1.90	0.01	5.37	0.02
cis-10-PPT	6.48	0.01	12.80	0.03	19.43	0.04
PyrPT	6.64	0.02	10.59	0.03	13.11	0.03
DPPT	8.53	0.02	10.74	0.03	13.90	0.03

**Table 4 antioxidants-11-01371-t004:** Values of the antioxidant activity of phenothiazine and its derivatives in 50% water–ethanol solution.

50% Water–Ethanol Solution	C1 = 0.01%		C2 = 0.1%		C3 = 1%	
	K, µM/min	Sr	K, µM/min	Sr	K, µM/min	Sr
PT	4.73	0.02	9.65	0.02	17.11	0.03
cis-10-PPT	12.19	0.03	17.84	0.03	24.93	0.03
PyrPT	15.65	0.03	19.47	0.03	24.93	0.03
DPPT	15.29	0.03	23.84	0.04	44.77	0.04

**Table 5 antioxidants-11-01371-t005:** Values of the antioxidant activity of phenothiazine and its derivatives in 70% water–ethanol solution.

70% Water–Ethanol Solution	C1 = 0.01%		C2 = 0.1%		C3 = 1%	
	K, µM/min	Sr	K, µM/min	Sr	K, µM/min	Sr
PT	6.69	0.02	15.26	0.03	20.69	0.06
cis-10-PPT	17.56	0.03	21.32	0.03	31.35	0.07
PyrPT	19.86	0.03	26.13	0.03	29.47	0.07
DPPT	25.50	0.05	38.46	0.07	71.27	0.08

**Table 6 antioxidants-11-01371-t006:** Values of the antioxidant activity of phenothiazine and its derivatives in 85% water–ethanol solution.

85% Water–Ethanol Solution	C1 = 0.01%		C2 = 0.1%		C3 = 1%	
	K, µM/min	Sr	K, µM/min	Sr	K, µM/min	Sr
PT	17.78	0.05	23.03	0.06	24.85	0.06
cis-10-PPT	27.36	0.06	35.11	0.07	39.90	0.07
PyrPT	22.57	0.06	29.87	0.07	33.06	0.07
DPPT	39.90	0.07	65.44	0.08	82.76	0.08

## Data Availability

Available upon request.

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
