# Peer review of "Antioxidant Properties of New Phenothiazine Derivatives"

_antioxidants, 2022, doi:10.3390/antiox11071371_

Round 1

Reviewer 1 Report

Manuscript ID: antioxidants-1799995

Title: “Antioxidant Properties of New Phenothiazine Derivatives”

General comments

In this study, the antioxidant activity of phenothiazine and its derivatives and the optimal conditions for the measurement to establish the relationship between the antioxidant activity of these substances and their concentration was evaluated.

The basic idea of the manuscript is good, and it could be of practical interest.

The manuscript is generally well written with a logic structure. Introduction summarized well the theoretical background of the research. Applied methods can be considered as adequate to investigate the research questions.

However, there are some explanations are missing.

Comments and suggestions:

- Comment on the prooxidant activity of the phenothiazine

- Discuss with previous work carried out with Phenothiazine Derivatives

- Lines 236-237. They would be better in material and methods

- How many times were the different analyses performed? Are the methods reproducible and repeatable?

- In discussion, the results are not compared with previous studies. Discuss with previous studies on antioxidant capacity of Phenothiazine derivatives

Author Response

Thank you very much for this useful comments, it really helps us to make manuscript better!

We improved our article according to recommendations. All changes marked in red here and in the text of improved manuscript.

Please consider this improved version for publication in “Antioxidants” journal.

Best regards,

Authors

Answers to questions:

Reviewer  Comments:

- Comment on the prooxidant activity of the phenothiazine

Answer: Note that the dual nature of phenothiazine exhibits both antioxidant and prooxidant properties, and this phenomenon is realized depending on the solvent or microenvironment in the biological object. Thus, in aqueous solutions, it exhibits mainly antioxidant properties, while in fats it could act as a pro-oxidant with the formation of a phenothiazinyl radical.

- Discuss with previous work carried out with Phenothiazine Derivatives

Answer: we add section about previous works dedicated different Phenothiazine Derivatives antioxidant properties and also enhance discussion with additional references.

- Lines 236-237. They would be better in material and methods

Answer: Yes, we have corrected it and transferred this paragraph to “material and methods”.

- How many times were the different analyses performed? Are the methods reproducible and repeatable?

Answer: we corrected this drawback. All measurement were conducted at least five times. We used here standardized voltametric methodology for antioxidant activity evaluation.

- In discussion, the results are not compared with previous studies. Discuss with previous studies on antioxidant capacity of Phenothiazine derivatives

Answer: The presented substances are being studied as antioxidants for the first time. However, we made comparisons with data from the parent phenothiazine. The results have been added to the discussion. We also include part about different derivatives antioxidant properties.

Thank You for comments and your time!

Reviewer 2 Report

The paper presents an investigation of the antioxidant activity of new phenothiazine derivatives by voltammetry. The influence of various factors on antioxidant activity was considered by the methods of experiment planning. The dependence of antioxidant activity on the molecular structure of the substances and the influence of different factors on their antioxidant activity was shown. The results permit consideration of these substances as candidates for further pharmacological investigations.

Some improvements of the presentation are required, as follows:

-        Please check language: some phrases are too long, logic is lost.

-       Indicate bibliographic references for antibacterial, antitumor and antituberculosis effects of phenothiazines (section 1 Introduction/page 1, range 38)

-          Pharmacological properties of the synthesized compounds are needed in Introduction

-         Correct the origin country of Phenothiazine powder purchased

-         Check the explaining of all the abbreviations used in the manuscript

-       Please explain why “It is important that non-linear dependences were observed not only at 85% ethanol in the background electrolyte, but also for other more dilute ethanol concentration”

Author Response

Thank you very much for useful comments, it really helps us to make manuscript better!

We improved our article according to recommendations. All changes were marked in red here and in the text of improved manuscript.

Please consider this improved version for publication in “Antioxidants” journal.

Best regards,

Authors

Answers to questions:

Reviewer comments:

Some improvements of the presentation are required, as follows:

-        Please check language: some phrases are too long, logic is lost.

Answer: We have corrected the language in the article; improved and simplified phrases in the text, when possible and mark changes in red.

-       Indicate bibliographic references for antibacterial, antitumor and antituberculosis effects of phenothiazines (section 1 Introduction/page 1, range 38)

Answer: we add additional literature review and appropriate references included in manuscript. We also enhance discussion part and include more literature sources.

-          Pharmacological properties of the synthesized compounds are needed in Introduction

Answer: we include part for it; however, the full assessment of pharmacological properties will require future in vivo test. It has not done completely yet.

-         Correct the origin country of Phenothiazine powder purchased

Answer: we have corrected this mistake.

-         Check the explaining of all the abbreviations used in the manuscript

Answer: we have corrected this point and spelling abbreviations when it appears for the first time.

-       Please explain why “It is important that non-linear dependences were observed not only at 85% ethanol in the background electrolyte, but also for other more dilute ethanol concentration”

Answer: We include explanation in the text. The nonlinear dependence in various ethanol solutions confirms the occurrence of a quasi-reversible process of oxygen electroreduction.

Thank You for comments and your time!

Round 2

Reviewer 1 Report

The majority of the comments have been emended in the revised manuscript.

Reviewer 2 Report

Dear Authors,

Thank you for considered the previous recommendations and make improvements of the first version of the manuscript.

I consider now that the manuscript can be published in this form.

I wish you succes in your future work!